# PeriTox-M, a Cell-Based Assay for Peripheral Neurotoxicity with Improved Sensitivity to Mitochondrial Inhibitors

**DOI:** 10.3390/cells14231929

**Published:** 2025-12-04

**Authors:** Anna-Katharina Holzer, Mira Dürr, Selina Multrus, Laura Dangel, Viktoria Magel, Marcel Leist

**Affiliations:** 1In Vitro Toxicology and Biomedicine, Department Inaugurated by the Doerenkamp-Zbinden Foundation, University of Konstanz, 78457 Konstanz, Germany; 2Center for Alternatives to Animal Testing in Europe (CAAT-Europe), University of Konstanz, 78457 Konstanz, Germany

**Keywords:** peripheral neurotoxicity, mitochondrial toxicity, metabolic switch, high-throughput toxicity screening

## Abstract

**Highlights:**

**What are the main findings?**
Peripheral neurons (PNs), cultured in galactose medium, were switched to a predominantly mitochondrial energy metabolism.PNs with an in vivo-like mitochondrial metabolism were more sensitive to toxicants.

**What are the implications of the main findings?**
The new PeriTox-M assay (in galactose medium) detects peripheral neurotoxicants that would be missed by other assays.The PeriTox-M assay has a higher sensitivity and specificity than the original PeriTox assay.

**Abstract:**

Human cell-based assays for neurotoxicity (NT) and developmental neurotoxicity (DNT) have reached a high level of readiness, but some tests require improvements in the specificity and sensitivity at which mitochondrial toxicants are detected. This study aimed to optimize the PeriTox assay, which uses peripheral neurons (PNs) and predicts the potential of chemicals to trigger peripheral neuropathies. By introducing a glucose-to-galactose switch in the medium composition, cells were forced to rely on mitochondrial respiration. Using pre-differentiated PNs cultured in either glucose (Glc) or galactose (Gal), we observed no major differences in baseline phenotype, gene expression, neurite outgrowth, or total ATP content. However, a marked metabolic shift was confirmed by the increased oxygen consumption in Gal conditions. Based on measurements of neurite growth and ATP levels, Gal-adapted neurons showed a heightened sensitivity, up to 7500-fold, to a range of mitochondrial respiratory chain (MRC) inhibitors. The sensitivity shift was high for inhibitors of MRC complexes I and III and modest or absent for unrelated compounds such as proteasome inhibitors or cytoskeletal poisons. For complex I-III inhibitors, the enhanced detection of mitochondrial neurotoxicants was coupled with a more accurate distinction between cytotoxic and neurite-specific effects, i.e., an improved assay specificity. In conclusion, our study on 39 compounds suggests that running the PeriTox assay in galactose increases its sensitivity and specificity for several mitochondrial toxicants, while no general disadvantages or shortcomings were observed. The modified version (PeriTox-M) may increase the performance of in vitro test batteries for scientific and regulatory applications.

## 1. Introduction

Neurotoxicity (NT) assays and developmental NT (DNT) test methods, based on human cells, may be optimized in their sensitivity by the choice of cell culture medium components. In the past, this has been exemplified for the NeuriTox (UKN4) assay as well as for the cMINC (UKN2) assay [1,2].

A comprehensive strategy for (D)NT testing, as exemplified by the DNT in vitro test battery (DNT IVB), includes a large number of assays based on different human neural cell types [3]. These include not only CNS neurons, astrocytes, and oligodendrocytes, but also peripheral neurons. The multiplicity of test systems, combined with various readouts, is meant to assess all potential disturbances of different key neurodevelopmental processes (KNDP) and physiological neuronal functions. This is required in order to cover most, or all, types of neurotoxicants.

For (D)NT testing on peripheral neurons, human induced pluripotent stem cells (hiPSC) can be differentiated into immature peripheral neurons (PNs) and frozen as aliquots for later use. After the thawing of the cells, PNs attach and start growing neurites of several micrometers in length (20–100 µm) in the first 24 h [4,5,6]. The PeriTox (UKN5) assay uses these PNs and their feature of rapidly growing neurites for (D)NT testing. After test substance exposure for 24 h, live cell staining allows for the image-based readout of cell viability and neurite area in parallel. This multiparametric approach enables discrimination between cytotoxicity and specific neurite toxicity, based on a validated prediction model [3,4,5,7,8].

Since neurons are highly energy-dependent cells, energy metabolism plays an important role in (D)NT. A continuous supply of ATP is needed for processes like axonal transport (especially in peripheral neurons), the maintenance of ion gradients and membrane potential, and the provision of cytoskeletal structures (which are especially important in developing neurons, as used in the PeriTox assay). Neurons, in particular their neurites and synapses, use predominantly mitochondrial substrates. Thus, they produce ATP mainly in mitochondria under most physiological conditions [9,10,11,12,13,14,15].

However, under standard culture conditions, cells exhibit a glycolytic phenotype rather than using mitochondrial respiration for ATP production, with only about 5% of the available glucose being oxidized in mitochondria [1,16,17]. Such a condition makes cells relatively insensitive to mitochondrial toxicants. This situation is further aggravated if cells are supplied with excessive amounts of glucose, as is common for many standard cell culture setups [16,18,19]. Thus, mitochondrial toxicants would easily be missed or underestimated under such conditions [1], even though an intact energy metabolism is highly important for PNs.

One solution to this problem is a change of the carbohydrate source in cell culture media. The replacement of glucose in the culture medium with galactose tunes the cellular metabolism from glycolysis to the predominant use of mitochondrial respiration, which has been demonstrated before in various cell types such as central neurons, hepatocytes, fibroblasts, chondrocytes, or muscle cells [1,20,21,22,23,24,25,26,27,28,29,30,31,32,33]. This not only enhances the sensitivity of the cells to mitotoxicants, but also mirrors the tissue-situation more effectively, as it switches the cells towards a predominant use of oxidative phosphorylation (mitochondrial ATP production).

Although a glucose–galactose-switch assay has already been established for NT testing in CNS neurons, i.e., LUHMES cells [1], the use of (human) peripheral neurons for NT testing is of great importance and should not be neglected. Some important toxicants are relatively selective for the peripheral nervous system, and a large fraction of the neurotoxicity observed in clinical practice affects the peripheral nervous system. This is particularly important for chemotherapy-induced peripheral neuropathies (CIPN) [34,35,36], where chemotherapeutic agents severely damage peripheral nerves, but show no signs of central neurotoxicity [37,38,39]. Thus, (D)NT testing on CNS neurons does not necessarily identify peripheral neurotoxicants and highlights the importance of testing on suitable cell systems [4]. Therefore, this study aimed at establishing a glucose–galactose-switch variant of the standard PeriTox (UKN5) assay. The goal was to thereby enhance the detection of mitochondrial neurotoxicants in peripheral neurons. While there are various types of mitotoxicants affecting, e.g., mitochondrial protein biosynthesis, mitochondrial DNA, or the urea cycle, the present study focused specifically on mitotoxicants affecting the electron transport chain (ETC), since this is the major effect known for many neurotoxicants.

## 2. Materials and Methods

### 2.1. Cell Culture Procedures

The human iPSC line Sigma iPSC0028 (EPITHELIAL-1, #IPSC0028, Merck, Darmstadt, Germany) was used for the differentiation of PNs. The cultures of iPSC were maintained under xeno-free conditions [40], exactly as described before [5]. The use of the cells was reviewed and accepted (17 June 2022) by the IRB of the University of Konstanz (IRB22KN006-03/w).

The differentiation procedure from human iPSC to PNs was performed as previously described [5]. In brief, neural induction was initiated by dual SMAD inhibition, using noggin (17.5 ng/mL) and SB-431642 (10 µM) (both from Bio-Techne, Minneapolis, MN, USA). The differentiation towards the peripheral neuron fate was directed using the small molecule inhibitors CHIR99021 (1.5 µM; Axon Medchem, Groningen, The Netherlands), SU5402 (5 µM; Bio-Techne, Minneapolis, MN, USA), and DAPT (5 µM; γ-secretase inhibitor IX; Merck, Darmstadt, Germany) [4,5,6,41]. After differentiation for 9 days, immature peripheral neurons were frozen in 90% fetal calf serum (FCS) (Thermo Fisher Scientific, Waltham, MA, USA) and 10% dimethyl sulfoxide (DMSO; Merck, Darmstadt, Germany).

The production of cultures for toxicity testing followed the previously published procedure [4,5,7] with the following adaptations: after thawing of the pre-differentiated cells, PNs were seeded at a density of 100,000 cells/cm^2^ on Matrigel^TM^ (Merck, Darmstadt, Germany) coated plates. Cells were cultured in either glucose-containing N2-S medium (N2-S-Glc; DMEM/F12, 1× GlutaMax [both from Thermo Fisher Scientific, Waltham, MA, USA], 0.1 mg/mL apotransferrin, 1.55 mg/mL glucose, 25 μg/mL insulin, 20 nM progesterone, 100 μM putrescine and 30 nM selenium [all from Merck, Darmstadt, Germany]) or in galactose-containing, glucose-free N2-S medium (N2-S Gal; SILAC^TM^ Advanced DMEM/F12 FLEX [Gibco/Fisher Scientific, Hampton, NH, USA], 15 μg/mL insulin, 400 μM GlutaMAX, 0.7 mM L-arginine, 0.5 mM L-lysine, 21.5 µM phenol red, 20 nM progesterone, 100 μM putrescine, 30 nM selenium, 92.5 μg/mL apotransferrin, 25 mM galactose) supplemented with CHIR99021 (1.5 µM), SU5402 (5 µM) and DAPT (5 µM). On DoD3, medium was changed to N2-S medium (Glc or Gal, respectively) supplemented with 12.5 ng/mL brain-derived neurotrophic factor (BDNF), 25 ng/mL glia-derived neurotrophic factor (GDNF), and 25 ng/mL nerve growth factor (NGF) (all from Bio-Techne, Minneapolis, MN, USA).

### 2.2. Assessment of Neurite Area and Cell Viability

Immature peripheral neurons were thawed and seeded at a density of 100,000 cells/cm^2^ in either N2-S Glc for the PeriTox assay (UKN5) or in N2-S Gal for the PeriTox-M assay (UKN5b). For toxicity assessment, cells were left to attach for 1 h at 37 °C, 5% CO_2_, followed by treatment with test compounds. The readout was performed after 24 h of compound exposure [4,5,7].

For the readout, neurons were stained with 1 µg/mL HOECHST-33342 (H-33342) and 1 µM calcein-AM (both from Merck, Darmstadt, Germany) one hour prior to the imaging. After incubation for 1 h at 37 °C, 5% CO_2_, images were acquired automatically using an ArrayScan VTI HCS microscope (Thermo Fisher Scientific, Waltham, MA, USA) [4,5]. Images were analyzed for neurite area and cell viability as previously described [42].

### 2.3. Interpretation of PeriTox(-M) Data

According to the validated PeriTox data interpretation procedure [3,43,44], test results allow classifying compounds as being non-specifically cytotoxic or specifically neurotoxic. For that purpose, benchmark concentrations (BMCs) are determined for the endpoints neurite area (NA) and cell viability (V). If their ratio [BMC_25_(V)/BMC_25_(NA)] is ≥3, the compound is classified as a neurite-specific neurotoxicant. Ratios < 3 indicate general cytotoxicity.

### 2.4. Immunofluorescence Staining

Protein expression was assessed qualitatively via immunofluorescence staining and microscopy. All samples were prepared and analyzed exactly as described before [5] using the primary antibody for βIII-tubulin (polymerized; mouse IgG1; BioLegend, San Diego, CA, USA; cat.# 921001) at a 1:1000 dilution, while F-actin was stained with fluorescently labeled phalloidin-555 (Molecular Probes, Eugene, OR, USA; cat.# 8953S) at a dilution of 1:500.

### 2.5. Transcriptome Data Generation and Analysis

Sample lysates were prepared and shipped to BioClavis (BioSpyder Tech., Glasgow, UK) as described before [5]. Measurement of the Human Whole Transcriptome v2.1 panel (including 22533 probes) was performed via the TempO-Seq targeted sequencing technology [45]. Note that the data are provided in the Supplementary Materials (as an Excel workbook), which have been deposited in the database Zenodo (zenodo.org, Version v1) under: DOI: https://doi.org/10.5281/zenodo.17457804 [46].

Gene expression data were processed and analyzed using the DESeq2 (v1.32.0) package in R (version 4.5.1) [47]. Raw read counts were first aggregated for genes with multiple measurements, and subsequently normalized by constructing a DESeqDataSet object utilizing the median-of-ratios normalization method. For quality control, count data were log-transformed using the transformation log_2_(count + 1). To identify outlier samples, the distribution of median expression values across all samples was examined, and the interquartile range (IQR) was computed. Samples exhibiting median expression values deviating by more than 1.5× IQR from the global median across all samples were identified as outliers and excluded from further analysis. Following this filtering step, a new DESeqDataSet was generated from the retained samples. Technical replicates were collapsed by summing raw counts per gene using the collapseReplicates function provided by DESeq2.

For downstream principal component analysis (PCA), variance stabilization transformation (vst) was applied to the filtered count data using the vst function from DESeq2. PCA was performed on the transformed data, using the 1000 genes exhibiting the highest variance across samples.

Differential gene expression analysis was performed relative to the “Glc 1 h” condition, which served as the reference baseline. For each sugar and time point combination, statistical significance was assessed using the Wald test as implemented in DESeq2. Genes were considered differentially expressed (DEGs) if they exhibited an adjusted *p*-value < 0.05, following Benjamini–Hochberg correction for multiple testing [48], and an absolute log_2_ fold change greater than 1.

For visualization, log_2_ fold change values were extracted across all contrasts relative to the baseline condition, and heatmaps were produced using the ComplexHeatmap R package (version 4.5.1) [49]. For visualization of absolute expression levels, the filtered count data was normalized as counts per million (CPM), and biological replicates were averaged per condition.

### 2.6. Seahorse Assessment of Mitochondrial Respiration

For experiments assessing the cellular oxygen consumption rate (OCR) [50,51,52,53] or the function of respiratory chain complexes [53,54,55], an Agilent Seahorse XF analyzer (Agilent, Santa Clara, CA, USA) was used. PNs were seeded at a density of 275,000 cells/well on Matrigel-coated Seahorse XF24 V7 PS cell culture microplates (Agilent, Santa Clara, CA, USA) in either N2-S Glc or N2-S Gal, supplemented with CHIR99021 (1.5 µM), SU5402 (5 µM), and DAPT (5 µM).

For the assessment of mitochondrial function, the XF Cell Mito Stress test was performed on intact PNs as described earlier [1,2,56]. The measured oxygen consumption rate (OCR) was always normalized to the respective cell count.

In order to assess mitochondrial respiratory chain (MRC) complex inhibition, PNs were permeabilized, complex-specific substrates and inhibitors were sequentially injected, and the OCR was measured as described earlier [1,25,57]. For normalization of the data, the last OCR measurement before test substance injection was used.

Detailed descriptions of both assays (for use with LUHMES cells) are published in the ToxTemp format in Annex A (A3, A4) of Alimohammadi et al. (2023) [56].

### 2.7. Intracellular ATP Levels

The intracellular ATP content was assessed in the plates used for the PeriTox(-M) assay. After the image analysis on a Cellomics ArrayScan VTI HCS microscope, cells were lysed within the plates. Then, the commercial reagent mix CellTiterGlo 2.0 (Promega, Madison, WI, USA), containing luciferase, was used to measure the ATP content luminometrically [58,59,60]. Data were normalized to the DMSO solvent control.

### 2.8. Data Handling and Statistics

If not mentioned otherwise, values are expressed as means ± SEM. If not indicated otherwise, experiments were performed at least three times (i.e., using three independent PN differentiations), with at least three technical replicates per condition. BMC_25_ values were calculated using the web-based tool BMCeasy (http://invitrotox.uni-konstanz.de/BMC/; accessed on 16 October 2025) [61]. Raw data have been deposited on Zenodo. They are publicly accessible under the following DOI: https://doi.org/10.5281/zenodo.17457804 [46]. Statistical procedures are described in the figure legends.

## 3. Results and Discussion

### 3.1. Characterization of Peripheral Neuron (PN) Populations Cultured in Glucose (Glc) and Galactose (Gal)

The effect of providing different carbohydrate sources to the PN population commonly used for the PeriTox (UKN5) assay was tested. The cells were pre-differentiated from hiPSC in large lots and frozen in aliquots to be used for starting the test runs (Appendix A) [5]. After thawing, the PNs were seeded, in parallel, in culture media that either contained glucose (Glc, 25 mM) or galactose (Gal, 25 mM) [1]. The exposure to the test compounds was for 24 h (Appendix A).

In initial experiments, we asked whether the change of the carbohydrate source altered the general phenotype of PNs. For this purpose, PNs cultured in Glc- or Gal-containing media were compared regarding their neurite outgrowth and ATP content. A qualitative evaluation of neurite and cell morphology based on the cytoskeletal structures βIII-tubulin and F-actin did not reveal significant differences between the two culture conditions (Figure 1A and Appendix A).

In the next step, the toxicity of typical (non-mitochondrial) peripheral neurotoxicants was compared in the Glc (standard) and Gal version of the PeriTox assay. Upon exposure to the chemotherapeutic drug taxol, a 25% decrease in the neurite area was observed at concentrations of about 2 nM, both in Glc and Gal conditions. The intracellular ATP levels and the general cell viability were not affected in this concentration range (Figure 1E,F and Appendix A). Also, for the microtubule depolymerizer colchicine, no significant difference was found for the BMC_25_ in Glc vs. Gal (Appendix A). The similar BMC_25_ values (within the normal range of assay variation) suggest that both versions of the PeriTox assay exhibit a similar sensitivity to non-mitochondrial toxicants.

Altogether, these initial studies indicate that neither cell viability, cell differentiation, nor its disturbance by non-mitochondrial toxicants are affected by the Glc → Gal exchange of the medium carbohydrate source.

### 3.2. Transcriptomics-Based Characterization of PN Populations Cultured in Glc and Gal

For further characterization of Glc- and Gal-cultured PNs, we used time-dependent transcriptome profiling. Expression levels for 19,000 genes were assessed at three time points after thawing (Supplementary Materials, ref. [46]). Data were obtained for the time points of test substance application (1 h) and assay readout (24 h) of the PeriTox assay. The third time point (96 h) corresponds to a later differentiation stage and was investigated in order to obtain an impression of whether long-term culture in Gal medium would lead to changes in the differentiation.

A principal component analysis (PCA) of the top 1000 variable genes provided an overview of the resulting data structure. Independent biological replicates, i.e., PN differentiations, of both conditions, Glc and Gal, clustered closely together. The first principal component coincided with increasing time of differentiation, while the carbohydrate source had no major impact on the cells’ development (Figure 2A). A more detailed analysis of genes characteristic of neurons, and specifically peripheral neurons, supported this observation (Figure 2B). High expression levels of (peripheral) neuronal marker genes such as *PRPH*, which encodes the peripheral neuron-specific intermediate filament peripherin [62], or *MAPT*, encoding the microtubule-associated protein tau, were already evident in Glc-cultured PNs one hour after thawing, and were not altered by cultivation in Gal. During the first 24 h in the respective media, the expression of neuronal marker genes remained largely stable and exhibited a high degree of similarity between Glc and Gal PNs. Until day 4 (96 h) after thawing, the transcriptional profiles of neuronal genes in both conditions progressed in a comparable manner. For instance, *PRPH* expression increased markedly, while *NEUROG1*, a marker of sensory neuron precursors [63,64], was downregulated.

Concurrently, genes associated with stem cell identity and the neural crest peripheral neuron precursor population, such as *KLF4* and *PAX3*, showed a time-dependent decrease in expression under both conditions. Thus, these data show that the Glc-Gal switch did not impact the development of PNs on the transcriptome level during the first 24–96 h after thawing. Furthermore, PN cultures used in the PeriTox assay under Glc- or Gal conditions exhibited high similarity in their overall transcriptome profile.

### 3.3. Mitochondrial Respiration of PNs Cultured in Glc and Gal

In order to verify that the use of Gal instead of Glc in the culture medium does induce a metabolic switch from a predominant use of glycolysis for energy production to mitochondrial respiration, the oxygen consumption rate (OCR) was measured as readout for mitochondrial respiratory activity (Figure 3A). The quantification of the basal respiration rate showed that PNs cultured in Gal consumed approximately twice as much oxygen as “Glc cells” (Figure 3B).

We used oligomycin A, an inhibitor of the mitochondrial complex V, also known as the ATP synthase, to quantify “ATP-linked respiration”. The latter parameter was significantly higher in “Gal PNs” than in PNs grown under standard Glc conditions. A subsequent injection of FCCP is known to uncouple oxygen consumption from ATP synthesis. Under these conditions, we observed that the maximal respiration that can be achieved by the mitochondria did not differ between Glc and Gal cells, but the spare respiratory capacity was higher in Glc cells. This is explained bioenergetically by Gal cells already running on their maximal respiratory capacity (Figure 3B and Appendix A).

Upon injection of the cI inhibitor rotenone and the cIII inhibitor antimycin A, the MRC was shut down completely, enabling the quantification of non-mitochondrial O_2_-consumption, which was the same for both culture conditions (Appendix A).

The characterization of the model system showed that changing the carbohydrate source from Glc to Gal induces a metabolic switch and increases the use of mitochondrial respiration while not altering the general phenotype and ATP content of PNs.

### 3.4. Sensitivity Shift in PNs Cultured in Gal for Two Well-Characterized MRC Inhibitors

In order to investigate whether the metabolic switch induced by the exchange of Glc to Gal in the cell culture medium had an effect on the cells’ sensitivity towards mitochondrial toxicants, two environmental toxicants (plant protection products) were chosen for pilot testing. We selected tebufenpyrad as a selective MRC inhibitor with extensive documentation that the Glc-Gal switch was successful for CNS-derived neurons [1,56]. Picoxystrobin also qualified as a well-characterized inhibitor, for which the switch worked for CNS-derived neurons and neural crest cells [1,2]. The concentration-response data for cell viability showed a clear (12 to 50-fold) shift in the sensitivity of cells to both inhibitors when the carbohydrate source was changed from Glc to Gal (Figure 4A and Appendix A). In parallel, the neurite growth was assessed. For tebufenpyrad, this endpoint was affected at 350-fold lower concentrations in Gal vs. Glc (Figure 4B and Appendix A). For picoxystrobin, the shift in sensitivity was 10-fold. Similar shifts as for neurites were observed when ATP levels were measured (Figure 4B,C and Appendix A).

The data on these two exemplary compounds show that running the PeriTox assay in Gal medium can strongly increase its sensitivity to mitochondrial toxicants. However, the degree of sensitivity shift may depend on the test compound, and possibly also on the endpoint assessed.

Concerning the classification of the compounds (test specificity), picoxystrobin was identified as a neurite-specific toxicant in both the Glc and Gal conditions. Tebufenpyrad was classified as cytotoxic in Glc (V/NA ratio = 1.6), but clearly neurotoxic (ratio = 7.1) in Gal. This highlights that the metabolic shift not only enhances the PNs’ sensitivity to MRC complex inhibitors in general, but also allows for a better discrimination between cytotoxicity and neurofunctional toxicity.

### 3.5. Effect of the Gal-Induced Metabolic Switch on PNs’ Sensitivity to Diverse Complex I Inhibitors

After the promising pilot experiments that showed an increased sensitivity to the complex I (cI) inhibitor tebufenpyrad, a group of seven additional cI inhibitors (rotenone, tolfenpyrad, fenpyroximate, deguelin, piericidine A, berberine, and MPP^+^) was tested. Concentration-response data were obtained for cell viability, neurite growth, intracellular ATP levels, and BMC_25_ values were calculated. The neurite area was affected at clearly lower concentrations in Gal medium than in Glc conditions. The shifts in sensitivity ranged from 3-fold (berberine) to 7500-fold (piericidine A) (Figure 5 and Appendix A). Cell viability was, for most compounds, also affected at lower concentrations in Gal medium. However, the shifts in sensitivity were smaller than those observed for neurites (Appendix A). The observation that neurite growth was affected more strongly than general cell viability is in line with the assay principle of the PeriTox assay. This test method is assumed to work for many neurotoxicants and especially peripheral neurotoxicants [4,5]. It is assumed that the long/growing neurites of PN cultures are particularly sensitive, as they have to fulfill specialized functions not found in many other (non-neuronal) cell types, e.g., axonal transport. In the in vivo situation, the sheer length and dependence on active transport processes are considered to be causal for the high sensitivity of PNs to many toxicants [65,66]. In our PN cultures, the growth process of neurites, requiring an extra transport and biosynthetic effort, may be the underlying sensitivity factor. In both cases, energy requirements play a role, and neurites in particular use mitochondrial energy substrates [9,13,14]. As the Gal medium makes them more dependent on mitochondrial function, this may explain the observed increase in neurite sensitivity.

Further support for this comes from our observation that intracellular ATP levels were affected by mitotoxicants at similar concentrations as neurite growth in Gal medium, while this was not necessarily the case in Glc medium, as shown, e.g., for rotenone (Figure 5A). This data set thus suggests a tight link between effects on ATP levels and neurite growth in cells that are predominantly using mitochondrial respiration for energy production. It thereby explains the increased sensitivity of Gal-PNs to MRC cI inhibitors.

### 3.6. Effect of the Gal-Induced Metabolic Switch on PNs’ Sensitivity to Diverse Complex II, III, and V Inhibitors, and Uncouplers

In order to test the hypothesis that Gal-PNs rely more on mitochondrial energy production and thus exhibit an enhanced sensitivity to toxicants interfering with mitochondrial ATP generation, we further tested a set of inhibitors and uncouplers of the MRC.

First, inhibitors of MRC complex II (cII; also known as succinate dehydrogenase in the citric acid cycle) were evaluated. Such compounds have given less clear results in other test systems than have cI inhibitors [1,25,67]. Possibly, a cII block can be bypassed more easily than a cI block. Moreover, most such compounds have been developed to block the fungal MRC, and they may be less specific for the human enzyme [67,68]. We observed here that a Glc-Gal switch increased the sensitivity to neurite damage for six (bixafen, isoflucypram, penthiopyrad, carboxine, thifluzamide, and 3-nitropropionic acid) of the nine tested compounds. However, the extent of the effect was moderate. For three of the tested compounds (atpenin A5, isopyrazam, and mepronil), no increased sensitivity was found (Figure 6 and Appendix A). Thus, the Glc-Gal switch did not affect all cII inhibitors, in contrast to its consistent and strong enhancement of the neurite toxicity of cI inhibitors. PNs in galactose medium appeared more sensitive to cII inhibitors than, e.g., LUHMES cells (CNS neurons) [1].

To further explore biological diversity and the potential applicability of the PeriTox assay in Gal medium, we next used a heterogeneous group of “other” mitochondrial inhibitors. Three uncouplers of the proton gradient were tested. A heterogeneous response pattern was obtained: neurite growth showed an increased sensitivity to 2,4-dinitrophenol in Gal medium. However, no effect difference was observed for the more potent uncoupler FCCP. Neither was there an altered sensitivity to the molluscicide tralopyril, a compound that acts as an uncoupler [69,70] (Figure 7A–C). We were also interested in compounds inhibiting the mitochondrial ATP synthase (cV). The inhibitor oligomycin A proved to be highly potent (BMC_25_(NA) = 1.5 nM) on PNs cultured in Gal medium, while no effect was observed in Glc medium at concentrations up to 100 nM (Figure 7D). We also tested 1-octylguanidine, which is a very low-affinity (10–100 µM range) cV inhibitor [71]. We did not observe a Glc → Gal sensitivity shift for this compound (Figure 7E).

As the cIII inhibitor picoxystrobin (Figure 4) showed promising results during our initial testing, we explored a panel of eight additional cIII inhibitors (antimycin A, fenamidone, pyraclostrobin, kresoxim-methyl, trifloxystrobin, fluoxastrobin, azoxystrobin, and cyazofamid). All cIII inhibitors applied in Gal medium affected the neurite area at concentrations at least threefold lower than when PNs were cultured in Glc medium (Figure 8 and Appendix A). Cell viability was also impacted with increased potency (>3-fold, except for cyazofamid) in Gal conditions (Appendix A).

Overall, the Glc → Gal switch not only led to an increased neurite sensitivity to MRC complex inhibitors, but it also increased the neurite specificity of the tested compounds. A good example is the cII inhibitor penthiopyrad: it was classified as cytotoxic (ratio V/NA = 1.3) in Glc medium, but induced neurite-specific effects (ratio V/NA = 6.0) in Gal medium.

For a selected subset of 10 mitochondrial inhibitors, we measured intracellular ATP levels to compare such data with the other endpoints. A consistent pattern emerged: ATP loss occurred at similar concentrations to those that impaired neurite outgrowth (Figure 6, Figure 7 and Figure 8 and Appendix A). This provides support to the hypothesis that ATP availability and neurite growth are closely coupled in neurons and that growing neurites depend on mitochondrial respiration for energy supply.

### 3.7. Effect of the Gal-Induced Metabolic Switch on PNs’ Sensitivity to Proteasome Inhibitors

Having characterized the PeriTox assay with a metabolic shift (to be named PeriTox-M or UKN5b) with a large panel of mechanistically well-characterized tool compounds, we performed a small case study on assay applicability. We chose proteasome inhibitors (PIs), as they do not have mitochondria as their primary target but are well known from clinical experience to trigger specific peripheral neuropathies [72,73,74,75]. For instance, bortezomib is a PI that is used as a chemotherapeutic drug to treat multiple myeloma. It causes peripheral neuropathies in up to 64% of treated patients [76,77,78,79]. It has been suggested that the compound’s side effects are related to an impaired mitochondrial function downstream of reduced proteasome activity [80,81,82,83]. For the PI MG-132, detailed experimental studies (metabolomics) indeed suggest such a downstream (indirect) inhibition of mitochondria [84]. On this basis, we chose a panel of five PIs (bortezomib, delanzomib, carfilzomib, epoxomicin, and MG-132) to investigate whether they would affect PN cultures with a higher potency after a Glc → Gal shift (Figure 9A–E). We found that the neurite growth of PNs was 4-fold more sensitive to bortezomib in Gal medium (with a BMC_25_ of 2.9 nM). Delanzomib, a peptide boronic acid PI that is structurally related to bortezomib, also affected the neurite area at lower concentrations in Gal medium. However, the sensitivity increase was only 2.5-fold. The PIs epoxomicin and carfilzomib (both irreversibly acting epoxyketones) showed only a slight increase in their potency (1.9 to 2.5-fold). MG-132 caused a clear shift of 3.2-fold. For all compounds, the drop in ATP levels went in parallel with effects on neurites in Gal medium (Appendix A). This was not due to a general effect on viability, which was only affected at >80-fold higher concentrations (except for epoxomycin). The concurrent reduction in ATP levels and neurite area suggests that impaired neurite outgrowth may be associated with an energy deficit. However, this effect was not attributable unambiguously to mitochondrial dysfunction, as cells cultured in glucose also exhibited a decline in intracellular ATP levels at similarly low PI concentrations.

In this application study of the PeriTox-M assay, the Gal-induced metabolic switch did not lead to clear (>3-fold) sensitivity shifts for the whole group of PIs. However, PNs cultured in Gal showed a moderately increased sensitivity to all tested PIs compared to Glc-cultured PNs.

Since bortezomib caused the most pronounced sensitivity shift, we verified for this compound that it did not interfere directly with the MRC of PNs: we adapted a previously established neuronal MRC complex inhibition assay to PNs, and we tested the cI inhibitors tebufenpyrad and fenpyroximate, the cII inhibitor carboxin, and the cIII inhibitors picoxystrobin and azoxystrobin as positive controls [53]. The assay results confirmed that the five MRC complex inhibitors specifically inhibited their respective target complex (Appendix A). Exposure to bortezomib, even at a concentration (1000 nM) exceeding its toxicity by >30-fold, did not reveal any direct inhibition of MRC complexes I–IV (Figure 9F).

While our findings speak against a direct effect of bortezomib on MRC complexes, a secondary impairment of mitochondria may contribute to PI toxicity, as suggested by a (moderately) higher potency in Gal medium of all tested PIs. These results suggest that performing the PeriTox test in a Gal-containing medium, as opposed to the standard glucose-based setup used to date, can generally increase sensitivity to neurotoxicants—even those that do not directly target mitochondrial complexes but may cause other types of mitochondrial damage.

## 4. Conclusions

We demonstrated here, conclusively, that a metabolic shift, converting the established PeriTox assay to the PeriTox-M assay (Gal-medium; mitochondria as main ATP producers), increases the test sensitivity without apparent disadvantages (e.g., compromised cell viability, altered phenotype, or loss of specificity). The main test readout, i.e., a specific neurite toxicity occurring at non-cytotoxic concentrations, was not impaired.

For an overall assessment of the performance of the PeriTox-M assay, we classified the tested compounds regarding their neurite specificity. The 39 exemplary toxicants tested in this study included inhibitors of the MRC complexes I-III and other types of MRC disruptors, as well as compounds that are known to primarily act on non-mitochondrial targets such as the proteasome. For all tested cI and cIII inhibitors, a clear Glc → Gal shift of >3-fold (and up to 7700-fold) indicated that their action on the MRC complexes had a stronger impact on the growth of neurites when PNs were cultured in Gal medium, i.e., relying primarily on mitochondria for energy production (Figure 10A,B). On this basis, all members of this toxicant subgroup were correctly classified as mitotoxicants. The Gal condition (PeriTox-M assay) led to an increase in the test specificity (Figure 10C and Appendix A).

Inhibitors of MRC cII have often been neglected in metabolic switch studies [22,25,27,28], or only small sets of compounds have been tested and the obtained data have been rather ambiguous [1,26]. Since cII is thought to play a more significant role in PNs than in CNS neurons [85], a set of nine cII inhibitors was included in our study. Not only was the sensitivity to cII inhibitors increased in Gal medium for all nine compounds compared to Glc medium, but also the specificity for neurite effects, i.e., the ratio of BMC_25_(V/NA), was increased throughout the panel of tested cII inhibitors: five of them were classified as being clearly neurotoxic in the PeriTox-M assay (Figure 10C and Appendix A). Comparing the results gained with PNs to previous results from CNS neurons, PNs were 2–9 times more sensitive to six of the assessed cII inhibitors (Appendix A).

This situation clearly differs from the one observed for cI inhibitors: CNS neurons were 5–50 times more sensitive to tebufenpyrad and tolfenpyrad than PNs [56]. This indicates that cI inhibition may have a higher impact on CNS neurons, while cII may play a greater role in PNs.

Another group of compounds that affect PNs in a significantly different way than CNS neurons are PIs, such as bortezomib. They are used as chemotherapeutic drugs, and one of their side effects is the induction of peripheral neuropathies. Bortezomib was shown earlier to be a specific neurotoxicant in the PeriTox assay [4,5] and in other human peripheral neuronal models [86], while it induced non-specific cytotoxicity in models of CNS neurons [1,87,88]. This specificity for neurite toxicity in PNs was further increased in the PeriTox-M assay. This was not only observed for bortezomib, but for all PIs tested (except for epoxomycin) (Figure 10C and Appendix A). Notably, bortezomib was non-specifically cytotoxic to CNS neurons in Gal medium in the same concentration range (Appendix A) [1]. Such differences between PNS and CNS neurons illustrate why it is important to develop test methods that cover different types of neurons and thus the different effects on them [3,89].

When assay performance is assessed, it is important to consider that an increased sensitivity does not necessarily lead to a higher accuracy. Specificity also needs to be considered. In this context, we found that toxicants with a known non-mitochondrial mode of action did not lose their specific neurite toxicity upon a Glc → Gal switch. For three types of toxicants (a microtubule disruptor, a blocker of microtubule dynamics, and several PIs), the PeriTox-M assay was not inferior to the classical PeriTox assay (Figure 10 and Appendix A). Specificity was not compromised, but rather enhanced. Thus, the new assay will also offer added value to testing batteries such as the DNT IVB [44] as a potential first-tier screening assay. Furthermore, the Glc → Gal switch could be explored to improve other neuronal test methods, based on, e.g., 3D cultures of peripheral or central neuronal organoids.

## Figures and Tables

**Figure 1 cells-14-01929-f001:**
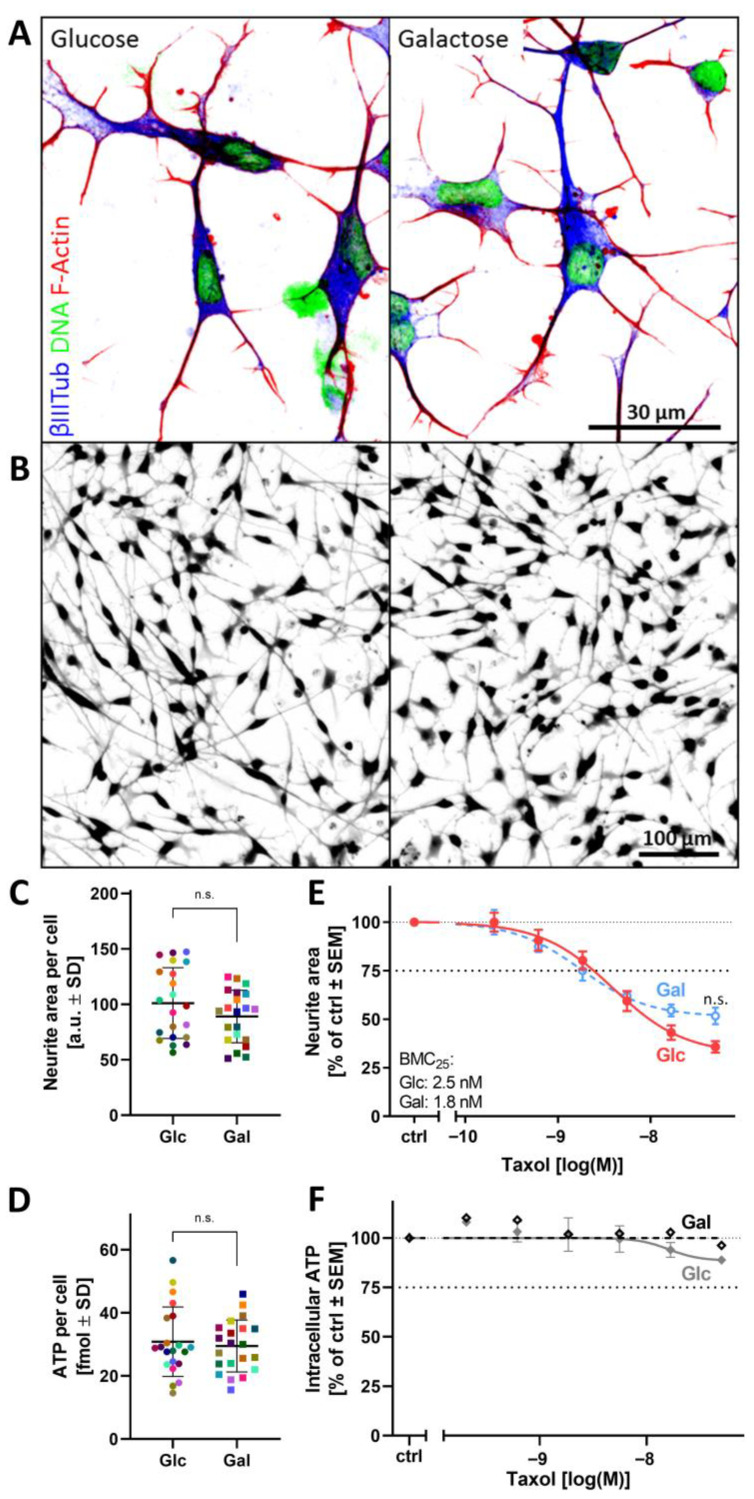
Basic characterization of peripheral neurons cultured in glucose or galactose media. Peripheral neurons (PNs) at the neurite outgrowth state were seeded in either glucose (Glc) or galactose (Gal) medium and characterized 24 h later. (**A**) PNs were fixed and stained for the neuronal cytoskeletal marker βIII-tubulin (βIIITub) and F-actin. DNA was stained with Hoechst-33342. The color code and scale bar are given in the images. (**B**,**C**) The neurite area was assessed via a calcein-AM/Hoechst-33342 live stain, and representative pictures of calcein-positive cells (**B**), as well as quantitative data on the neurite area (**C**), are given. (**D**) For the 21 independent experiments evaluated in (**C**), the ATP content was measured and normalized for cell number. Color-matched data points are derived from the same biological replicate, squares represent data in galactose, and dots data in glucose conditions. The means ± SD are indicated. (**E**,**F**) PNs cultured in Glc (solid) or Gal (dashed) were exposed to taxol for 24 h: (**E**) The neurite area and (**F**) the ATP content were assessed. The benchmark concentration at which a 25% decrease was observed (BMC_25_) is given in the graph. Data are means ± SEM of 3–5 biological replicates. Differences were tested for statistical significance using an unpaired, parametric *t*-test with Welch’s correction. n.s., not significant. More supporting data are visualized in Appendix A. The full data set is given in Supplementary Materials, ref. [46]. The overall appearance of the cell culture (cell density, cell morphology, neurite structures) also did not show obvious differences (Figure 1B). Since the neurite area is the main functional endpoint of the PeriTox assay, quantitative data were acquired on this endpoint in 21 independent experiments: there was no difference between PNs cultured in either Gal or Glc (Figure 1C). Moreover, intracellular ATP levels (assessed after 24 h) were similar in the two conditions (Figure 1D).

**Figure 2 cells-14-01929-f002:**
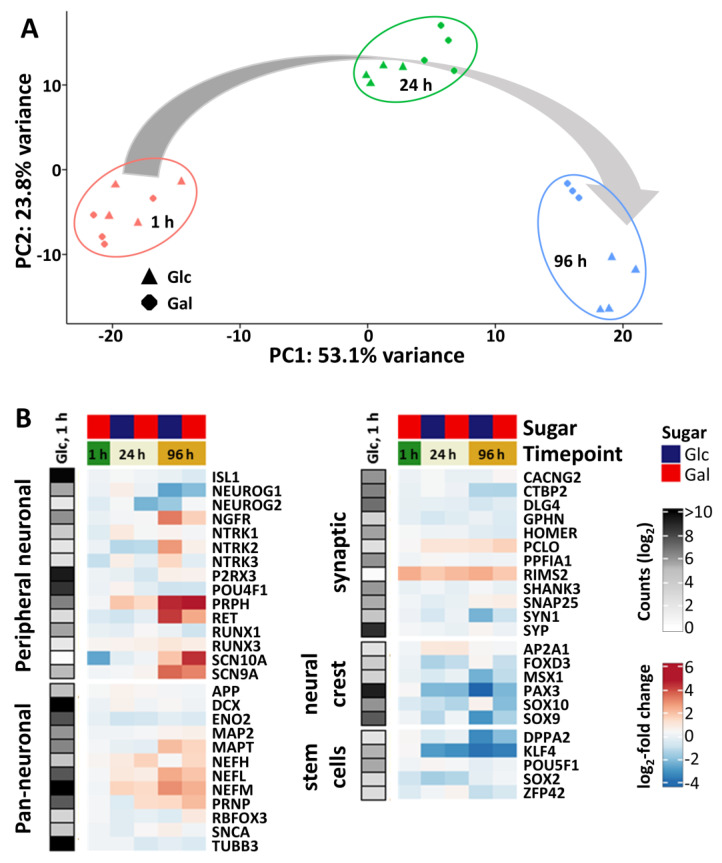
Transcriptome profiling of iPSC-derived PNs cultured in glucose or galactose. TempO-Seq transcriptome analysis was performed for three differentiation stages (1 h, 24 h, and 96 h after thawing) of PNs cultured in glucose (Glc) or galactose (Gal) medium (full data in Supplementary Materials, ref. [46]). (**A**) For the top 1000 variable genes of this data set a principal component analysis (PCA) was performed. In the 2-dimensional PCA display of principal components (PC) 1 and 2, three differentiation stages of PNs are color-coded. Samples from Glc-cultured cells are shown as triangles, Gal samples as diamonds. Each data point represents an independent differentiation of PNs with three technical replicates for each. (**B**) Genes of interest, clustered into five groups (peripheral neuronal, pan-neuronal, synaptic, neural crest, and stem cells), were evaluated for their expression levels over time. The left column (b/w) illustrates the absolute expression levels in counts of the corresponding gene for cells cultured in Glc for 1 h after thawing. The color scale uses log_2_(counts per million), ranging from white (no expression) to black (high expression). Changes in the gene expression levels (given as log_2_-fold change) over time are visualized for Gal and Glc samples relative to the expression of the reference sample (Glc, 1 h). The color scale ranges from blue colors (decreased expression levels) via white (no changes) to red colors (increased gene expression). Data are means of 4 independent differentiations with three technical replicates each.

**Figure 3 cells-14-01929-f003:**
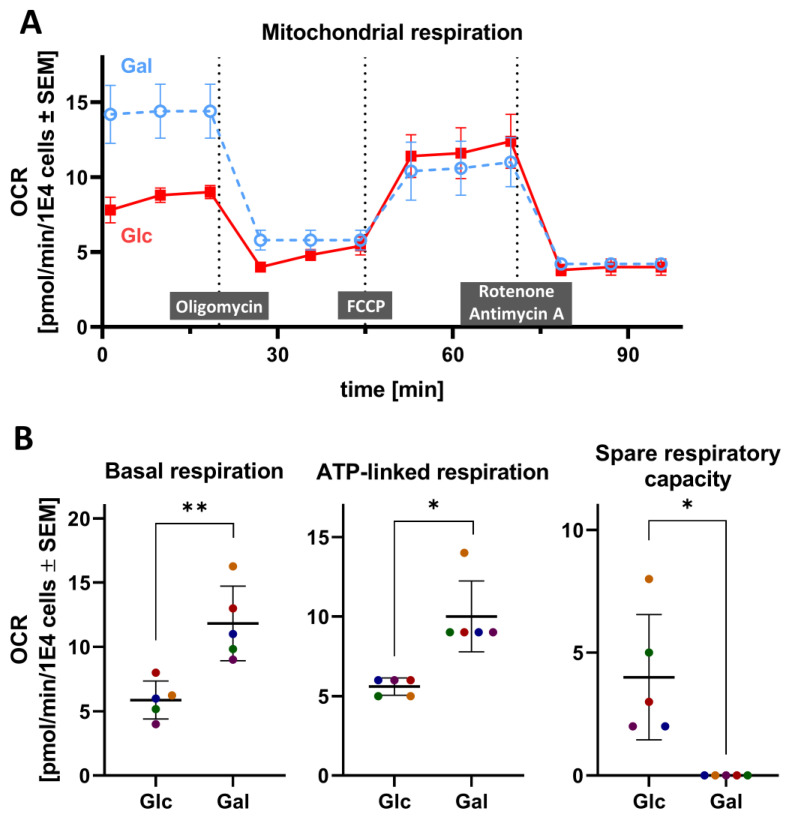
Comparison of mitochondrial respiration parameters in glucose vs. galactose cultured peripheral neurons. Peripheral neurons were seeded in either glucose (Glc) or galactose (Gal) media as in Appendix A. Mitochondrial respiration was investigated after 24 h. (**A**) The oxygen consumption rate (OCR) was measured for various respiratory states: cells were sequentially treated at the indicated time points (vertical dotted lines) with the complex V inhibitor oligomycin A, the uncoupler FCCP, and a combination of the complex I inhibitor rotenone and the complex III inhibitor antimycin A. Following each inhibitor addition, the OCR was measured for 20 min. (**B**) From the OCR data, the basal respiration, ATP-linked respiration, and the spare respiratory capacity were calculated. Color-matched data points are derived from the same experiment, different colors refer to data from different biological replicates. Differences were tested for statistical significance using an unpaired, parametric *t*-test with Welch’s correction. Data are means ± SEM of 5 independent experiments. * *p* < 0.05; ** *p* < 0.01.

**Figure 4 cells-14-01929-f004:**
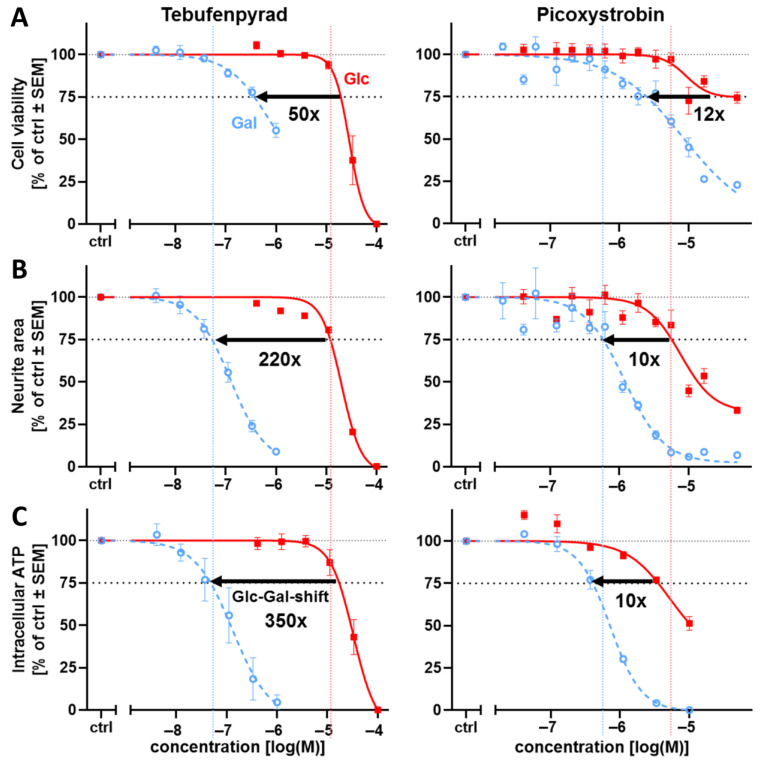
Enhanced sensitivity of PNs for the complex I inhibitor tebufenpyrad and the complex III inhibitor picoxystrobin. PNs were cultured as in Appendix A in either glucose (Glc, red, solid) or galactose (Gal, blue, dashed). After attachment, the cells were treated for 24 h with the mitochondrial respiratory chain (MRC) complex I inhibitor tebufenpyrad (left) or the MRC complex III inhibitor picoxystrobin (right). (**A**) The cell viability was assessed via calcein-AM/Hoechst-33342 live staining, and the relative number of viable cells (normalized to the total cell count) is given. Note that the highest test concentration under Gal conditions for tebufenpyrad was 1 µM (sufficient to derive a BMC_25_ value). (**B**) Based on the same high-content imaging data, the total neurite area was determined. (**C**) In parallel, the ATP content was measured in the same assay plate. (**A**–**C**) The black arrow indicates the offset of the BMC_25_ caused by switching from Glc to Gal (see data overview in Appendix A). The number below shows the ratio of the BMC_25_ detected in Glc medium vs. Gal medium. Vertical lines that correspond to the BMC_25_ (neurite area) are depicted to allow better visual comparison of the curves. Data points are given in percent of the control group (ctrl, 0.1% DMSO) and are means of 3–5 independent biological replicates ± SEM (see Supplementary Materials, ref. [46], for full data set).

**Figure 5 cells-14-01929-f005:**
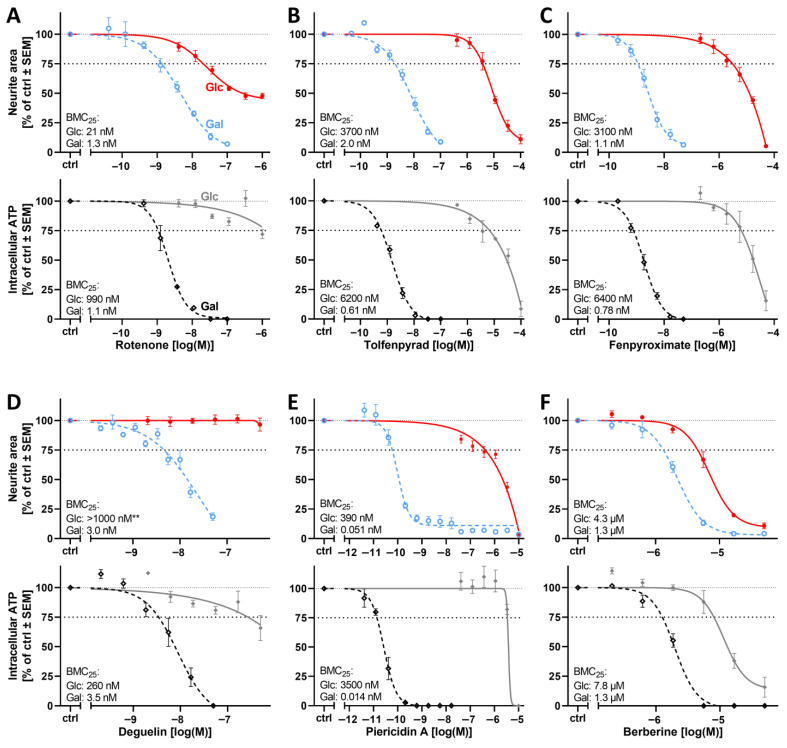
Effect of the glucose–galactose switch on the peripheral neurotoxicity of mitochondrial respiratory chain complex I inhibitors. Various inhibitors of the mitochondrial respiratory chain complex I (cI) were tested for their neurotoxic potential. PNs were cultured in either glucose (Glc, solid) or galactose (Gal, dashed), as in Figure 4. After attachment, the cells were treated for 24 h with cI inhibitors (**A**–**F**). The neurite area was assessed via calcein-AM/Hoechst-33342 live staining (upper graphs, colored). In parallel, the ATP content was measured on the same assay plate (lower graphs, grays). The BMC_25_ is given for each condition in the respective graph. Respective viability data are given in Appendix A. Data points are given in percent of the control group (ctrl, 0.1% DMSO) and are means of 3–5 independent biological replicates ± SEM (see Supplementary Materials, ref. [46], for full data set). ** indicates that no effect (<10%) was measured. The dashed and dotted lines have been inserted for better orientation to indicate 100% and 75% values.

**Figure 6 cells-14-01929-f006:**
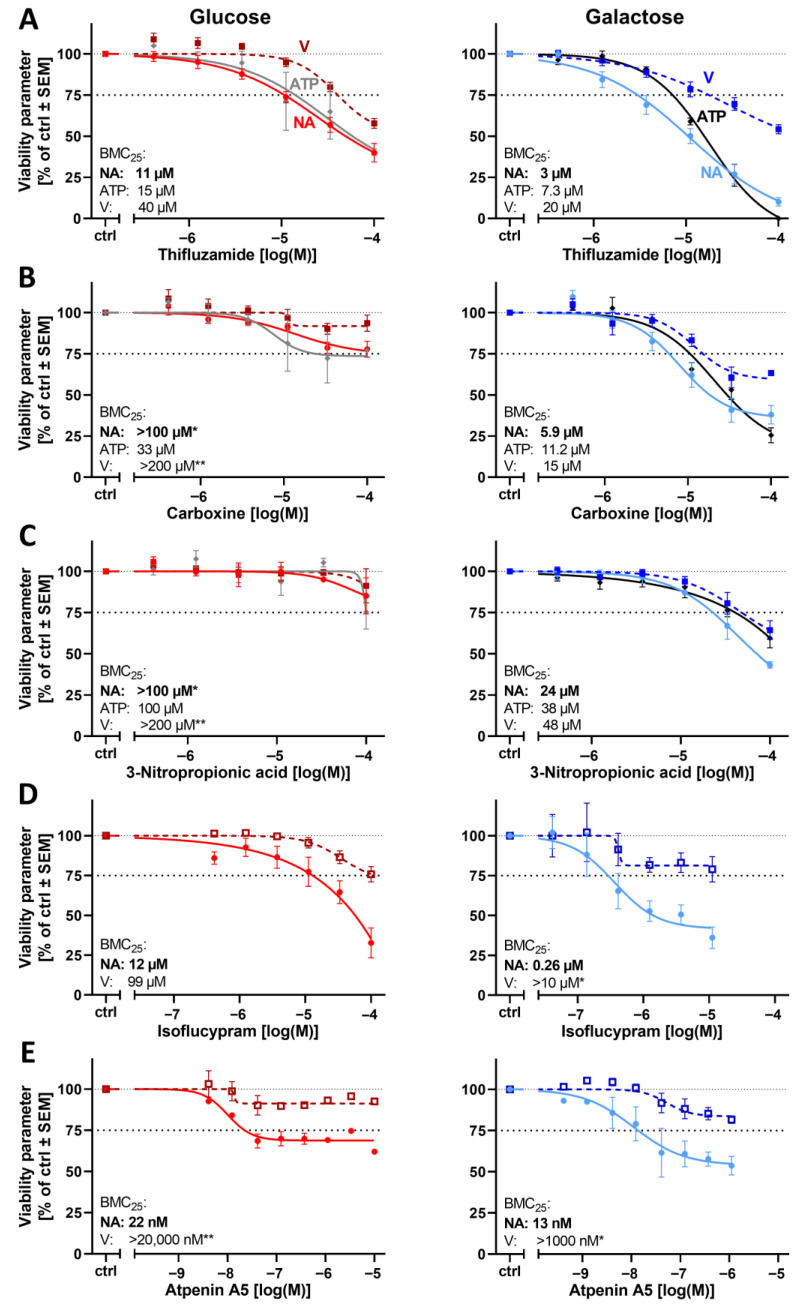
Effect of the glucose–galactose switch on the peripheral neurotoxicity of mitochondrial respiratory chain complex II inhibitors. Various inhibitors of the mitochondrial respiratory chain complex II (cII) were tested for their neurotoxic potential. PNs were tested in either glucose (left) or galactose (right) as in Figure 4. After attachment, the cells were treated for 24 h with cII inhibitors (**A**–**E**). The cell viability (V) was assessed after calcein-AM/Hoechst-33342 live staining by high content imaging. The relative number of viable cells (normalized to the total cell count; squares, dashed line) and the total neurite area (NA; circles, solid line) were determined. In parallel, the ATP content was measured on the same assay plate (diamonds, solid line). The BMC_25_ is given for each condition and endpoint in the respective graphs. Data points are given in percent of the control group (ctrl, 0.1% DMSO) and are means of 3–5 independent biological replicates ± SEM (see Supplementary Materials, ref. [46], for full data set). * indicates that the BMC_25_ could not be determined in the tested concentration range, but at least a 10% effect was measured. ** indicates that no effect (<10%) was measured. The dashed and dotted lines have been inserted for better orientation to indicate 100% and 75% values.

**Figure 7 cells-14-01929-f007:**
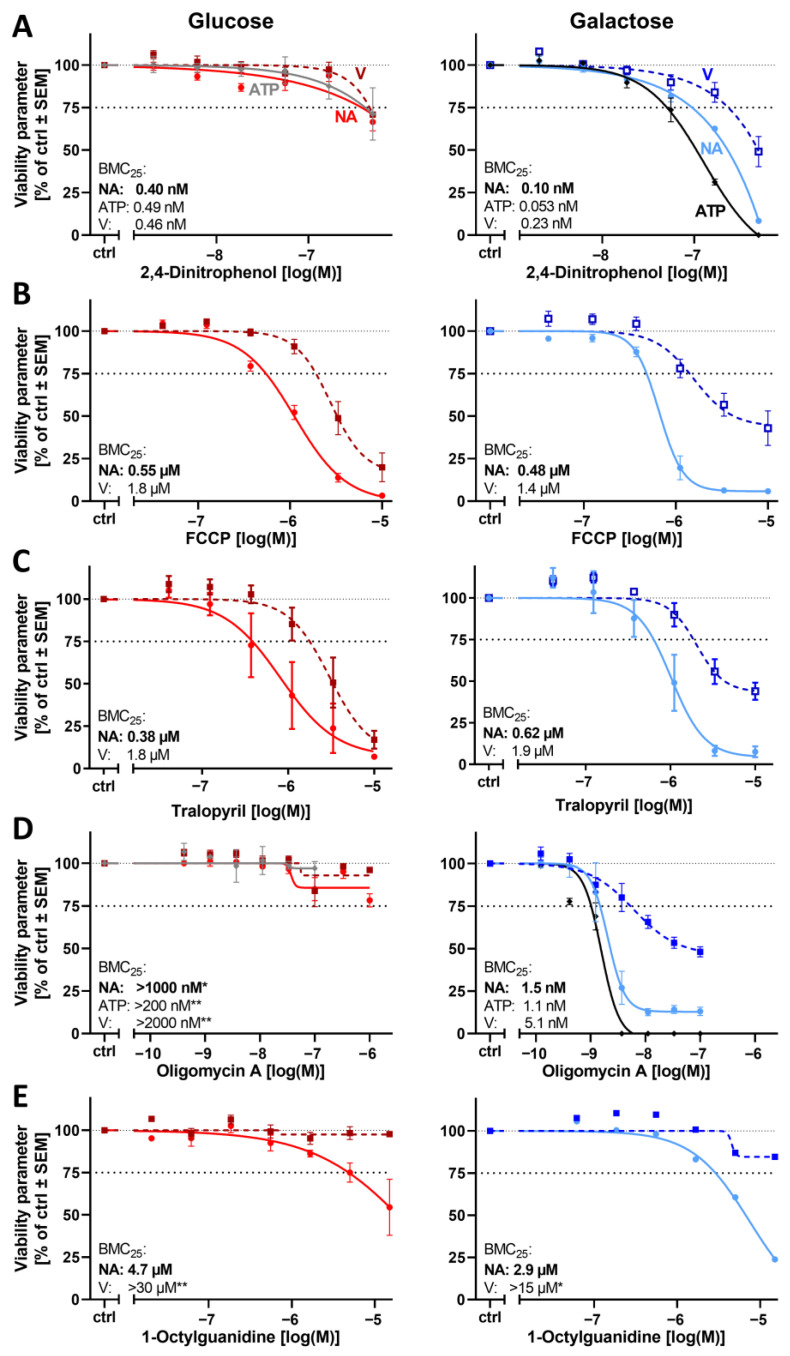
Effect of the glucose–galactose switch on the peripheral neurotoxicity of mitochondrial respiratory chain ATPase inhibitors and uncouplers. Inhibitors of the mitochondrial respiratory chain (MRC) complex V (cV, ATPase) and MRC uncouplers were tested for their neurotoxic potential. PNs were thawed and seeded in either glucose (Glc, left) or galactose (Gal, right) as in Figure 4. The cells were treated for 24 h with the uncouplers (**A**–**C**) and cV inhibitors (**D**,**E**). Cell viability (V; squares, dashed line), the total neurite area (NA; circles, solid line), and the ATP content (diamonds, solid line) were measured. The BMC_25_ is given for each condition and endpoint. Data points are given in percent of the control group (ctrl, 0.1% DMSO) and are means of 3–5 independent biological replicates ± SEM (see Supplementary Materials, ref. [46], for full data set). * indicates that the BMC_25_ could not be determined in the tested concentration range, but at least a 10% effect was measured. ** indicates that no effect (<10%) was measured. The dashed and dotted lines have been inserted for better orientation to indicate 100% and 75% values.

**Figure 8 cells-14-01929-f008:**
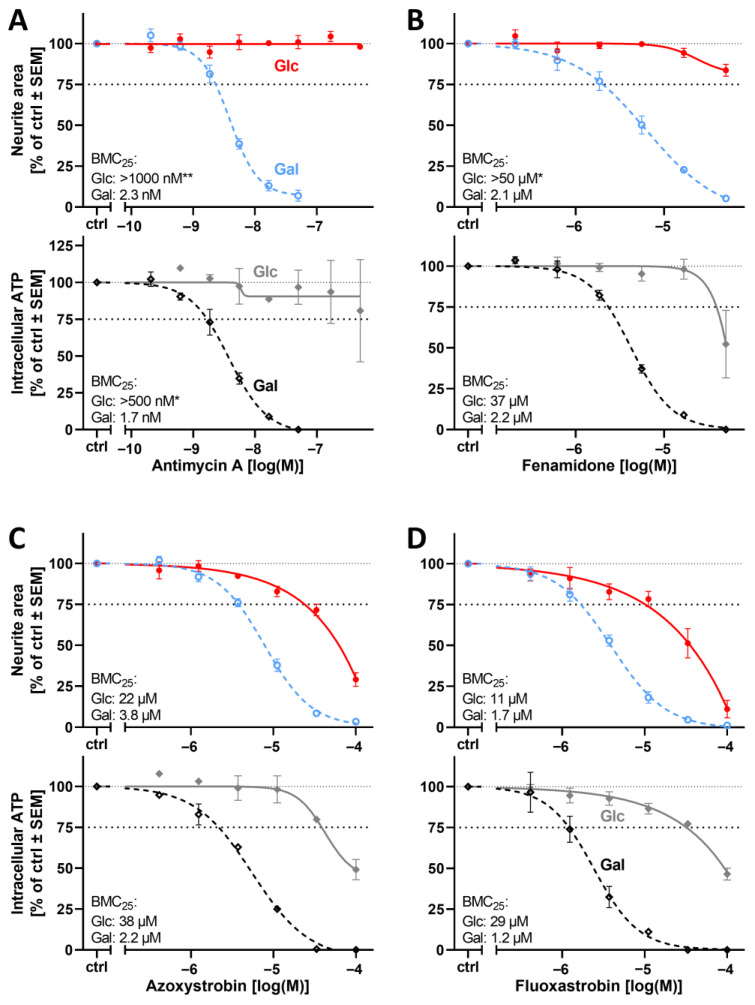
Effect of the glucose–galactose switch on the peripheral neurotoxicity of mitochondrial respiratory chain complex III inhibitors. Various inhibitors of the mitochondrial respiratory chain complex III were tested for their neurotoxic potential, as in Figure 4. Cells were treated in glucose (Glc) or galactose (Gal) for 24 h with cIII inhibitors (**A**–**D**). The neurite area (**upper** graphs, colored) and the ATP content (**lower** graphs, grays) were measured on the same assay plate. The BMC_25_ is given for each condition and endpoint in the respective graph. Respective viability data are given in Appendix A. Data points are given in percent of the control group (ctrl, 0.1% DMSO) and are means of 3–5 independent biological replicates ± SEM (see Supplementary Materials, ref. [46], for full data set). * indicates that the BMC_25_ could not be determined in the tested concentration range, but at least a 10% effect was measured. ** indicates that no effect (<10%) was measured. The dashed and dotted lines have been inserted for better orientation to indicate 100% and 75% values.

**Figure 9 cells-14-01929-f009:**
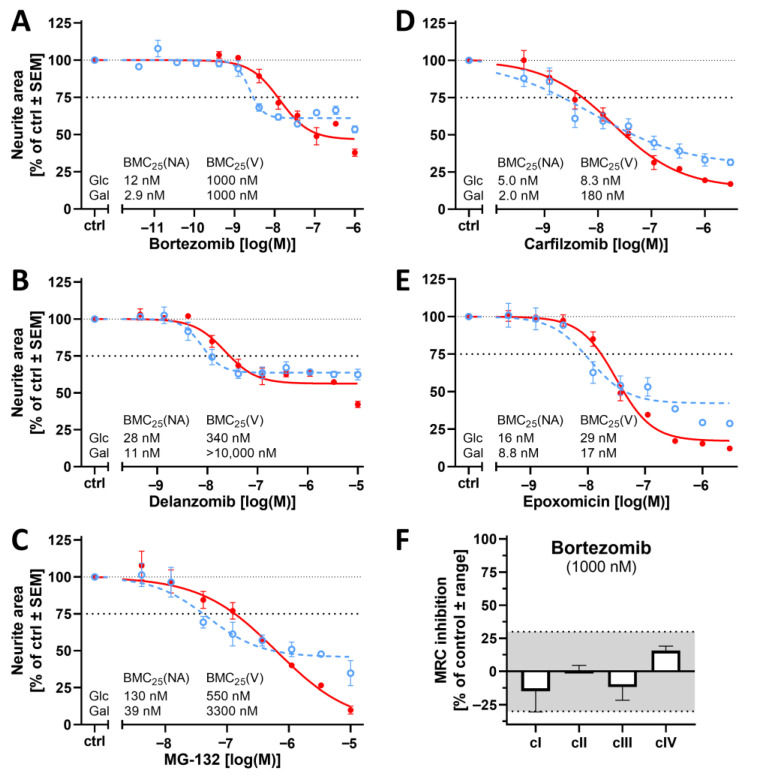
Effect of the glucose–galactose switch on the peripheral neurotoxicity of proteasome inhibitors. Various proteasome inhibitors were tested for their neurotoxic potential. PNs were cultured in either glucose (Glc, solid) or galactose (Gal, dashed) as in Figure 4. The cells were treated for 24 h with proteasome inhibitors (**A**–**E**). The neurite area was assessed via calcein-AM/Hoechst-33342 live staining and is depicted as concentration-response curves. The BMC_25_ is given for each condition. Data points are given in percent of the control group (ctrl, 0.1% DMSO) and are means of 3–5 independent biological replicates ± SEM (see Supplementary Materials, ref. [46], for full data set). The dashed and dotted lines have been inserted for better orientation to indicate 100% and 75% values. (**F**) PNs were used in a mitochondrial respiratory chain (MRC) complex inhibition assay to assess potential direct effects of bortezomib on single MRC complexes. PNs at the neurite outgrowth state were seeded in glucose medium. After 24 h, cells and their outer mitochondrial membranes (but not the inner mitochondrial membrane) were permeabilized. Their oxygen consumption rate (OCR) was measured in the presence of specific substrates (electron donors) for the single complexes and complex-specific inhibitors after acute injection of bortezomib at a concentration of 1000 nM. The non-significant range of the assay was defined by 2× SD of the baseline variation and is indicated by the light gray area. Data are means ± range of two replicates and are given as % inhibition compared to the OCR of control cells (0% inhibition).

**Figure 10 cells-14-01929-f010:**
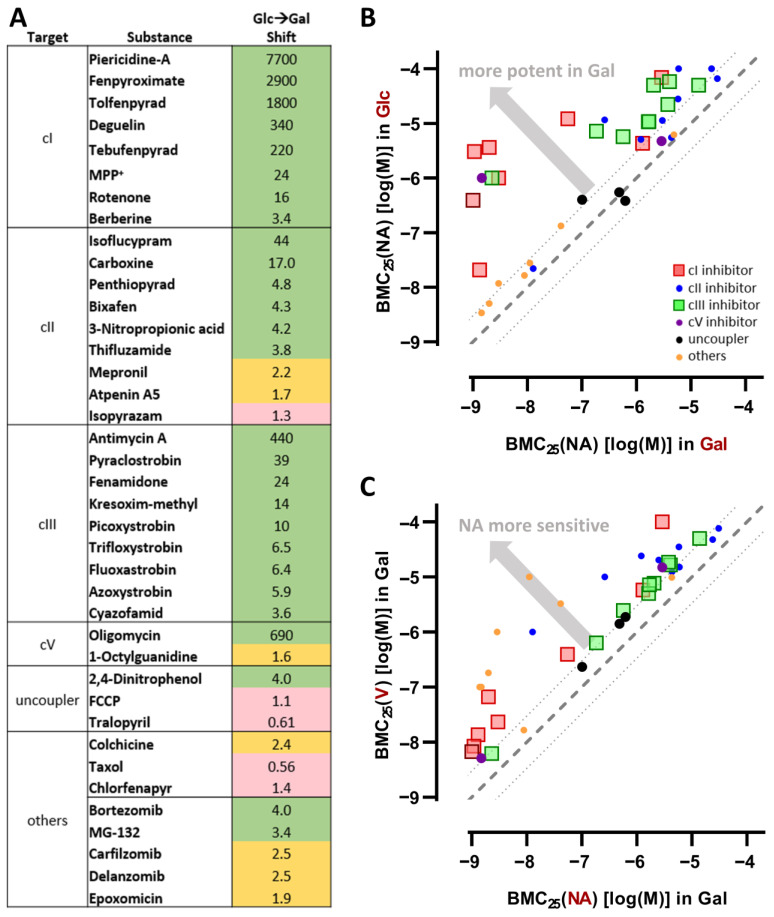
Comprehensive overview of tested substances revealing increased sensitivity of the PeriTox-M assay without loss of specificity. The PeriTox-M assay was evaluated regarding its sensitivity to the 39 test compounds used in this study and its specificity in detecting neurotoxic effects. (**A**) For all test compounds, the Glc → Gal shift, i.e., the ratio of the BMC_25_(NA) in glucose (Glc) and the BMC_25_(NA) in galactose (Gal), was calculated. This value gives information about the increase in sensitivity when tested in the PeriTox-M assay. Following the classification by Delp et al. (2019) [1], cells were assigned colors based on the increase in sensitivity by galactose: red for ratios below 1.5 (no enhancement), yellow for ratios between 1.5 and 3 (moderate enhancement), and green for ratios above 3 (strong enhancement). (**B**) This enhancement in sensitivity is displayed graphically for all tested compounds. The BMC_25_(NA) in Gal is given on the *x*-axis, and the BMC_25_(NA) in Glc on the *y*-axis. The gray dashed line indicates a BMC_25_(NA) [Glc/Gal] ratio of 1, while the dotted lines indicate a ratio of 3 (and 0.33). Every compound found left of the upper dotted line was detected in the PeriTox-M assay with strongly increased sensitivity, indicated by the gray arrow. (**C**) The PeriTox-M assay’s capability to detect specific neurotoxic effects is illustrated graphically across all tested compounds. The BMC_25_(NA) in Gal is given on the *x*-axis, and the BMC_25_(V) in Gal on the *y*-axis. The gray dashed line indicates a BMC_25_(V)/BMC_25_(NA) ratio of 1, while the dotted lines indicate a ratio of 3 (and 0.33). Every compound found left of the upper dotted line decreased the neurite area (NA) more potently than the general cell viability. Such compounds were considered specific neurotoxicants. (**B**,**C**) The different compound groups are assigned to shapes and colors. Detailed data are found in Supplementary Materials, ref. [46].

## Data Availability

Raw data have been deposited on Zenodo and are publicly accessible under the following DOI: https://doi.org/10.5281/zenodo.17457804 [46].

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
