# Peer review of "PeriTox-M, a Cell-Based Assay for Peripheral Neurotoxicity with Improved Sensitivity to Mitochondrial Inhibitors"

_cells, 2025, doi:10.3390/cells14231929_

Round 1

Reviewer 1 Report

Comments and Suggestions for Authors

This is an excellent manuscript that has well-constructed and well-controlled experiments, clear, outstanding illustrations, and meaningful results. The strategy and methodology are easy to follow.  In the end, the authors provide a useful and sensitive tool for studying mitochondrial neurotoxicity, which frequently contributes to neurodegeneration. Minor clarifications or corrections would further improve the manuscript.

  • Does the Glc--> Gal adaptation work for primary neuronal or slice culture models?
  • How is the approach or data interpretation impacted by ER stress?
  • Why was the Gal-Tebufenpyrad dose-effect curve truncated relative to Glc (Fig 4A)?

Author Response

Comment 1: This is an excellent manuscript that has well-constructed and well-controlled experiments, clear, outstanding illustrations, and meaningful results. The strategy and methodology are easy to follow.  In the end, the authors provide a useful and sensitive tool for studying mitochondrial neurotoxicity, which frequently contributes to neurodegeneration.

Response 1: We thank the reviewer for the positive feedback and appreciate the recognition of our work.

Comment 2: Minor clarifications or corrections would further improve the manuscript. Does the Glc--> Gal adaptation work for primary neuronal or slice culture models?

Response 2: We agree that this is an interesting point. It is not answered in the current manuscript for three reasons, but will be addressed in the future (see below). (i) In the current study, we do not only focus on peripheral neurons (iPSC-derived) as test system, but in particular on a specific toxicological test method using these cells (the PeriTox, UKN5 assay). A major part of the work was to run this test method under Glc and Gal conditions and to compare the outcome. The comparison of the untreated cell cultures as such was a comparatively minor part of the study. To do the same for primary cells, one would need to choose defined toxicological assays based on such cells and run the whole library of compounds for comparison. (ii) The question may be less pressing or relevant for primary cells, in particular brain slices. The reason is that cell lines tend to adapt to their culture conditions by decreasing mitochondrial ATP generation and increasing glycolytic flux. Primary cells, in particular neurons, have (at least initially) a more pronounced mitochondrial usage. They do not need to be forced to use their mitochondria to the same extent as cell lines. (iii) In many cases, the technical problem of removing glucose is very high. Some media are offered as glucose-free variant. In such cases, glucose or galactose can be added. For some media this option is not given. This also includes some media used for primary cells. We are currently working on a solution for this. It will allow any type of cell or any type of medium to be used. We will prepare a full manuscript on this.

Comment 3: How is the approach or data interpretation impacted by ER stress?

Response 3: We thank the reviewer for highlighting this important topic. From the point of view of cell biology, there are several links between ER stress and mito stress, e.g. via calcium handling, ER-mito contact sites and similar downstream responses, such as ATF4 activation. We focused here initially on mitochondrial respiratory chain inhibitors, as they are known to be “under-detected”, and as there are dozens of such compounds, which are relevant for human exposure as environmental toxicants. This situation is less clear for ER stressors. We are actually interested in following this idea, starting with model compounds, such as thapsigargin and tunicamycin, moving on to toxicologically more relevant compounds (HSP inhibitors, brefeldin etc..), and moving on to some compounds that induce such stress indirectly (several plant ingredients). We have worked for a while on identifying specific measures of ER stress. Often ER stress is assessed by its effect on protein folding: elements of the unfolded protein response (UPR) are used as a readout. The biomarker most frequently measured in this context is ATF4 or one of its target genes. We found earlier, that MPP+ and MG-132, both included in the present study (but not classified as direct ER stress inducers) trigger a very prominent and rapid ATF4 response. Moreover, the ATF4 response is also triggered by nutrient (amino acid) depletion. An overarching problem is that the UPR is a general readout of toxicity, also in liver, kidney and other markers. And many unspecific toxicants indirectly lead to ER and mitochondrial stress. For this reason, a single endpoint cannot be specific and a complex panel of endpoints is required to clearly identify compounds that directly induce ER stress (and to distinguish them from other stress inducers). We want to invest some more work into it, before we use data for publication.

Comment 4: Why was the Gal-Tebufenpyrad dose-effect curve truncated relative to Glc (Fig 4A)?

Response 4: We thank the reviewer for pointing out this source of confusion. The ending of the curve is not really a “truncation”. Moreover, this ending at a point above the baseline is no problem, as our assay’s main readout is the BMC25 value. We are quite experienced in concentration-response modeling and we have published also software for this. During this work we observed that the right part of the curve contains very little relevant information for determining toxicological thresholds. Therefore, our curve fitting algorithm strongly focuses on the left part of the curves. Of course, we use pre-screening experiments to identify the relevant test ranges (not shown). Based on this, we defined a maximal test concentration under Gal-conditions to be 1 µM in Fig. 4A. The curves are thus not “truncated”. They all end at 1 µM (highest tested concentration). This is a common procedure to ensure specificity of data. For instance, in Fig. 5A, the highest test concentration was 0.1 µM. Also there, no information beyond this point was required to derive BMC25 values, i.e. a concentration that leads to a drop from 100% to 75%. We have now added an explanation to the figure legend of Fig. 4.

Reviewer 2 Report

Comments and Suggestions for Authors
  1. I feel that in the introduction, the authors should introduce more why they focused on peripheral neurons, not other neurons or glial cells. I have a feeling that peripheral neurons are not the widely studied cells in neurotoxicity studies. So, additional explanations may be needed to explain why investigating neurotoxicity in peripheral neurons is important.
  2. Did the authors characterize peripheral neurons for their markers? How do they know that they successfully obtain peripheral neurons?
  3. The authors stated that the established methods have no apparent disadvantages in abstract and conclusions. This may be too much. This study only tested one type of cell, and it may be unknown for other cell types. I hope that they authors could still discuss about the limitations, such as the need to test on other cell types or organoids.

Author Response

Comment 1: I feel that in the introduction, the authors should introduce more why they focused on peripheral neurons, not other neurons or glial cells. I have a feeling that peripheral neurons are not the widely studied cells in neurotoxicity studies. So, additional explanations may be needed to explain why investigating neurotoxicity in peripheral neurons is important.

Response 1: We agree with the reviewer concerning a neuroscience/cell biology perspective. However, from the point of view of toxicology and assay development, our approach seems justified. We now combined this in following paragraph in the introduction: “Some important toxicants are relatively selective for the peripheral nervous system and a large fraction of the neurotoxicity observed in clinical practice affects the peripheral nervous system. This is particularly important for chemotherapy-induced peripheral neuropathies (CIPN) [34-36], where chemotherapeutic agents severely damage peripheral nerves, but show no signs of central neurotoxicity [37-39]. Thus, (D)NT testing on CNS neurons does not necessarily identify peripheral neurotoxicants and highlights the importance of testing on suitable cell systems [4].” We would like to stress that the Glc-Gal assay has already been developed for central neurons (indicated in the introduction), but is urgently required for peripheral testing. Peripheral neurotoxicity is of overwhelming importance in many areas. The problem is that only few model systems are available and they can be reproducibly mastered only in few laboratories.

Comment 2: Did the authors characterize peripheral neurons for their markers? How do they know that they successfully obtain peripheral neurons?

Response 2: We agree with the reviewer that a detailed characterization is very important; in particular, for a cell system like peripheral neurons that is difficult to generate. The basic characterization (morphology, marker expression, gene expression, functional responses) was the topic of our previous work e.g. Hoelting et al., 2016 and Holzer et al., 2022, cited in the manuscript. This work showed, that the cells closely resemble human dorsal root ganglion cells and express all the relevant markers to be considered peripheral neurons. We avoided here repetition of such basic work, however, we show in Fig. 2 the expression of main peripheral marker genes and how those are stable over multiple days of culturing; most importantly, they are not changed by a shift in the carbohydrate source (Glc/Gal).

Comment 3: The authors stated that the established methods have no apparent disadvantages in abstract and conclusions. This may be too much. This study only tested one type of cell, and it may be unknown for other cell types. I hope that the authors could still discuss about the limitations, such as the need to test on other cell types or organoids.

Response 3: We would like to clarify the misunderstanding about our claims. We fully agree with the reviewer that claiming that a carbohydrate shift has no disadvantages for any cell type would be going too far and would be too speculative. Our paper focused specifically on one cell type (iPSC-derived peripheral neurons) applied for one specific toxicological test. This is what our statement referred to. We would not even claim that the “no disadvantage” is true for other assays using such cells. For instance, we also use peripheral neurons for studies on pain signaling. For this, a 50-day differentiation is required. We have not explored how the shift of Glc to Gal would affect this differentiation, and thus we would never claim any effects or no-effects, we have not measured. This notwithstanding, we would like to mention that the Glc-Gal shift assay has already been established for LUHMES cells (cells of the central nervous system). Furthermore, we mentioned in our introduction that this system has also already been established for other cell types like hepatocytes, fibroblasts, chondrocytes, neural crest cells or muscle cells. Nevertheless, testing this assay system on other neuronal cell types like glial cells or astrocytes, or on other culture systems like neuronal organoids are interesting research topics for the future. Preliminary testing on peripheral neuronal organoids in our lab, under glucose and galactose conditions has revealed a higher sensitivity to mitochondrial inhibitors under galactose conditions as well. We refer to this in the outlook paragraph (in very generalized form) in our conclusions section.